

# Factors limiting reproductive success in urban Greylag Geese (*Anser anser*)

Sabrina Mai[1,2], Jean-Louis Berthoud[3], Holger Haag[1] and Friederike Woog[1]

[1] Department of Zoology, State Museum of Natural History Stuttgart, Stuttgart, Germany
[2] Center of Excellence for Biodiversity and Integrative Taxonomy, University of Hohenheim, Stuttgart, Germany
[3] Unaffiliated, Neuchâtel, Switzerland

## ABSTRACT

In the late eighties, Greylag Geese (*Anser anser*) started to colonise an urban area previously void of geese in southwestern Germany. Between 2004 and 2020, in a period of steady population increase with subsequent population stagnation, we analysed two measures of reproductive success: (1) the relation between freshly hatched to fledged young for each brood and (2) the probability of a hatchling to survive to fledging. We were able to show that the dispersal of pairs from the nesting site to a different brood rearing area resulted in higher reproductive success. However, the increasing population size of Greylag Geese and the number of breeding pairs of recently immigrated Egyptian Geese (*Alopochen aegyptiaca*) had a negative impact on reproductive success, indicating density dependence. Our results show that newly established populations in urban settings do not grow indefinitely, which is an important fact that should be taken into account by wildlife managers.

## INTRODUCTION

In the Anthropocene, understanding population limitations in animals is becoming increasingly important (*Pelletier & Coltman, 2018*). While the majority of species are declining or even disappearing, some profit from the presence of humans. As their populations increase, conflicts may result in calls for hunting and population regulation. Geese are a prime example, as most populations in the western Palaearctic have increased considerably in the last century (*Fox & Leafloor, 2018*) due to a shift from natural to agricultural or urban habitats, where they damage crops or soil recreational spaces (*Lowney et al., 1997*; *Fox & Abraham, 2017*; *Fox et al., 2017*). Understanding the parameters that drive population changes in wild animals remains a prerequisite for their management. A crucial parameter is reproductive success (*Siriwardena, Baillie & Wilson, 1998*). It is influenced by a wide range of biotic and abiotic factors, which include the intrinsic quality of an individual itself (such as the ability to find nutritious food, a good mate and nesting site, its age and breeding experience, and its ability to cope with intra- and interspecific competition) but also environmental factors such as food availability, predation, or weather conditions (*Newton, 1998*; *Angelier et al., 2007*). When populations increase, particularly

Corresponding author
Sabrina Mai,
sabrina.mai@smns-bw.de

in a defined space, intraspecific competition may increase and density dependence may occur. The primary mechanism behind this is resource competition, particularly for space and food (*Newton, 1998*). As a result, the growth rate of a population decreases at higher densities (*Lande et al., 2002*). This increasing density of a population can have an effect on reproductive success, for example through increased nest predation as shown in Savannah sparrows *Passerculus sandwichensis* (*Woodworth et al., 2017*).

There are many measures of reproductive success and they may vary across species. In birds, these include clutch size, number of hatchlings or fledglings, surviving number of offspring into the next year or percent of offspring in a population (*Howard, 1979*; *Clutton-Brock, 1988*; *Owen & Black, 1989*). Ecologically, the different measures are indicative of selective forces active at different times of the annual cycle and locations of a bird. For example, in geese, clutch size depends on the nutritional status of the female (*Ankney & MacInnes, 1978*). The number of hatchlings can be indicative of the defensive abilities at the nest (*Szipl et al., 2019*) and the number of fledglings may additionally vary with food supply and predation pressure (*Williams et al., 1993*; *Fondell et al., 2008*). Likewise, the survival of offspring into the next year and the percent of offspring in a population may also depend on dangers faced during migration, such as hunting (*Clausen et al., 2017*).

Among the goose populations which have been increasing over the last decades is the well-studied Svalbard population of migratory Barnacle Geese *Branta leucopsis*, which went from a low of 300 individuals estimated in the 1940s to over 12,000 individuals in the 1980s (*Prestrud, Black & Owen, 1989*) and continued to increase at a higher rate to an estimated 45,000 individuals in 2015 (*Fox & Leafloor, 2018*). The initial population growth led to higher numbers of potential breeders and in turn to an increase in the number of nests (*Prestrud, Black & Owen, 1989*). However, nest sites were limited and by the 1980s, most of the breeding area was occupied (*Owen, 1984*; *Prop, van Eerden & Drent, 1984*). The overall reproductive success of the population (defined as the percent of young in the autumn flock in the wintering grounds in Scotland) decreased with increasing population size (*Owen & Black, 1989*). During this time, the availability of food near the nesting sites on Svalbard influenced reproductive success, as food scarcity near the nesting sites led to parents feeding further away and leaving nests unattended (*Prop, van Eerden & Drent, 1984*). The low availability of food also led to feeding competition between families during post-hatching. This may have reduced the fledglings' ability to store reserves for migration and thus resulted in a decreasing number of juveniles in the wintering quarters (*Owen, 1984*). To mitigate this, several goose species move considerable distances between breeding and post-hatching areas, such as Lesser Snow Geese *Anser caerulescens caerulescens* in Manitoba, Canada. Between 1973 and 1987, the population size nearly doubled (*Cooch et al., 1989*) and continued to increase thereafter (*Fox & Leafloor, 2018*). Because of subsequent overcrowding at the nest sites, feeding conditions became poor and led to a dispersal of families with freshly hatched young to the surrounding areas of up to 50 km. Goslings from pairs which dispersed from the breeding area had a higher pre-fledging survival and this resulted in a higher reproductive success of these pairs (*Cooch et al., 1993*). This dispersal refers to the temporary movement of breeding pairs with their

brood prior to fledging. It is important to distinguish it from natal dispersal, which is a permanent movement after fledging (*Cooch et al., 1993*).

*Lack (1947)* hypothesized that reproductive success in birds would be highest at an optimum clutch size. This optimum would be smaller than the largest possible clutch size (*Lessells, 1986*). Two hypotheses consider this optimum predicted by Lack: in one, the parents' reproductive success would be weighed against their survival into following years, while in the other, the resources invested into breeding would be weighed against the amount of necessary care (*Davies, Krebs & West, 2012*). In geese, an in-depth study looked at Lesser Snow Geese in Manitoba, where larger clutches had a higher rate of survival (*i.e.,* a higher fitness) than the smaller ones, but the effect was nearly linear (see *Rockwell, Findlay & Cooke, 1987* for an overview). In birds with long-term pair bonds, reproductive success is also affected by the choice of the partner and the breeding experience with that partner: In Svalbard Barnacle Geese, the reproductive success of pairs increased with the duration of the pair bond (*Black, Choudhury & Owen, 1996*; *Choudhury & Black, 1994*), while it decreased in Eurasian Oystercatchers *Haematopus ostralegus* after a divorce (*Ens, Safriel & Harris, 1993*). In Black-browed Albatross *Thallasarche melanophris*, fledging success increased with age and breeding experience (*Angelier et al., 2007*).

In this paper, we study a local population (as defined by *Wells & Richmond, 1995*) of introduced, urban Greylag Geese *Anser anser* in the city of Stuttgart, southwestern Germany. A ringing project started in 2002 allowing individual identification. Subsequently, data on local population size and individual reproductive success has been collected over a period of 17 years (2004–2020). In 2010, the first brood of Egyptian Geese *Alopochen aegyptiaca* successfully fledged in the same area, adding a new element to the local avifauna. Both species share the same feeding grounds and brood rearing areas. We therefore also monitored the local population size and number of breeding pairs of Egyptian Geese.

Within our defined study area, (1) we describe the population size and the number of breeding pairs of Greylag and Egyptian Geese over a period of 40 years. In the more detailed data set beginning in the year 2004, we analysed several hypotheses regarding reproductive success of ringed Greylag Geese. We use two measures of reproductive success: firstly, fledging success, *i.e.,* the relation between freshly hatched to fledged young for each brood, and secondly, hatchling survival, *i.e.,* the probability of an individual hatchling to survive to fledging. By analysing fledging success, we are able to look at reproductive success on the level of breeding pairs. This measure of reproductive success may vary with the parents' abilities. Hatchling survival specifically analyses the likelihood of each individual gosling to survive to fledging. There, parental influence is only one of the several parameters which may influence the gosling's survival.

We hypothesize that the fledging success of individual goose pairs (2) decreases due to density dependence, (3) increases with breeding experience and (4) will be higher for pairs which bred more often when accounting for all broods of a pair. (5) Additionally, we analyse fledging success across brood sizes to determine the optimum brood size for our local population.

On an individual level, we hypothesize that hatchling survival (6) decreases with the following variables or a combination thereof due to density-dependent mechanisms: (a)

an increasing local population size; (b) the number of breeding pairs in a population; (c) dispersal of pairs with young goslings to a new brood rearing area; (d) a higher number of hatchlings per pair or (e) the increasing number of Egyptian Geese and their breeding pairs due to competition or agonistic behaviour.

## METHODS

### Study area

Our study area is part of the city of Stuttgart, the capital of Baden-Württemberg in southwest Germany. The herbivorous geese use the urban parks for feeding and roosting, including the Inner City Parks (48°47′54″N 9°12′24″E) near the city centre of Stuttgart and a recreational area approximately 4 km to the north, the Max-Eyth Lake (48°50′03″N 9°12′55″E) (*Käßmann & Woog, 2008*). Within the Inner City Parks, several small ponds provide water refuges. Additionally, the river Neckar is nearby and connects to the Max-Eyth Lake.

### Study population

In Germany, there are both autochthonous and introduced Greylag Goose populations, with the latter being classified as regionally established (*Bauer et al., 2016*). The population in Stuttgart is such an introduced population. The first successful brood was reported in 1995 at the Max-Eyth Lake (*Woog, Schmolz & Lachenmaier, 2008*). Initially, the local population increased only very slowly, but by 2020, it had grown to an average of 300 individuals (*Hohmann & Woog, 2021a*). A large proportion of the birds (over 95%) are ringed with a metal ring of the "Vogelwarte Radolfzell" on one leg and a blue "Darvic" ring with white writing on the other leg ($n = 1252$; 2002–2020), allowing individual identification. Ring recoveries indicate that the local Greylag Geese are mostly non-migratory and stay in the area all year round, even during cold winters (*Käßmann & Woog, 2007*).

Greylag Geese prefer breeding sites close to the water and on the ground, ideally out of reach from terrestrial predators (*Hart & Downs, 2020*; *Groom et al., 2020*). Greylag Geese have average clutch sizes of five to six eggs, though in rare cases exceptionally large broods of 14 eggs are possible (*Glutz von Blotzheim, 1990*). In Stuttgart, they nest almost exclusively on three islands at the Max-Eyth Lake. Over the years, more and more geese had to share this very limited breeding space, leading to higher nest densities. While many geese stay within the city year-round, some also fly to locations outside of Stuttgart after moult. Among these, many non-breeders return from outside the city for moulting from mid-May to mid-June. Some geese also breed outside the city limits and return to the city with their fledged young (*Woog, Maierhofer & Haag, 2011*).

The first brood of Egyptian Geese occurred in 2010 and since then, the local population has been growing (*Hohmann & Woog, 2021a*). While the two goose species share the same feeding habitats, Egyptian Geese utilize more diverse nesting sites, often in tall trees, and have larger clutches than Greylag Geese, averaging eight to nine eggs per clutch (*Cramp et al., 1978*; *Callaghan & Brooks, 2016*; *Huysentruyt et al., 2020*). As breeding sites for Greylag Geese are limited in the study area, Egyptian Geese have a breeding advantage (*van Dijk,*

*2015*). Although known to be aggressive towards smaller water birds, Egyptian geese rarely attack Greylag Geese (*Hohmann & Woog, 2021b*).

## Animal welfare

No animals were harmed as a consequence of the study. Permissions to catch and ring geese were obtained from the Regierungspräsidien Stuttgart and Karlsruhe (AZ 55-8853.17/S; 55-9213.47 and 55-8841.03).

## Data collection

Starting in 1985, counts of Greylag Geese in the defined study area were carried out through opportunistic observations of unmarked birds. From 2002 to 2006, weekly counts of mostly ringed individuals were conducted by a well-trained group of observers. From 2007 onwards, weekly counts (including unringed geese) were carried out by the same observer (H. Haag). During the weekly counts of Greylag Geese, the local population size and number of breeding pairs of Egyptian Geese were also monitored. The term "local population size" refers to the highest count of geese during moult in the study area.

Access to the nest sites of Greylag Geese was not possible, as the islands are a Special Protected Area according to the EU's Birds Directive and are also used by a breeding colony of Grey Herons *Ardea cinerea* which may not be disturbed. Therefore, it was not possible to count the eggs of individual clutches but only freshly hatched goslings in the weekly counts. Mortality in freshly hatched goslings is highest in the first days of life, so opportunistic observations of birders in the area increased data quality during the first week after hatching.

As most Greylag Geese pair for life (*Lorenz, 1992*), we used the fledging success (in percent) of pairs rather than individuals. Each pair seen with goslings was therefore given a unique pair-ID. Pairs which bred over several years retained the same pair-ID throughout the years. When pairs separated and one of them bred with a new partner, a new pair-ID was given.

Non-breeders were defined as those geese which were seen during at least two weekly counts in the period of 15th March to 15th April, thus being present in Stuttgart during the breeding season without having bred themselves. After the breeding season, many geese from the surrounding areas moult in Stuttgart, but these were not counted as non-breeders as they may have bred elsewhere. They were, however, included in the highest annual count used as local population size.

## Statistics

All data analyses were conducted in R 3.6.2 (*R Core Team, 2019*) run from R Studio (*RStudio Team, 2021*) using the lme4 package (*Bates et al., 2015*) and lmerTest package (*Kuznetsova, Brockhoff & Christensen, 2017*). Brood sizes were defined as the initial number of young after hatching, as the number of eggs could not be counted. Only broods that hatched were included in the analyses ($n = 237$ broods with 938 hatchlings in total). We analysed two measures of reproductive success: fledging success and hatchling survival.

Fledging success was defined as the relation between freshly hatched to fledged young for each brood. Fledging was defined as the moment when the goslings were able to fly.

The percentage of fledging success per brood (number of fledglings compared to number of hatchlings) was analysed with a linear model. For each brood, the number of times a pair had bred previously was calculated using the IDs of the pairs and the year as reference, resulting in a discrete value for each brood symbolizing the nth time a pair bred. The effect of the nth brood by a pair on fledging success was analysed across all pairs using a mixed-effect model (LMM). The ID of each pair and the year of observation (2004–2020) were added as random effects to control for pseudo-replication. Subsequently, the relation between the mean fledging success of all broods by a pair and the total number of broods was analysed in an LMM, controlling for the pair-ID and year. In both cases, the best fit was chosen using post-regression residual analysis and mechanistic reasoning (*Godfray, Partridge & Harvey, 1991*; *Black & Owen, 1995*).

The impact of brood size on fledging success was analysed using an LMM controlling for the pair-ID and year. Post-regression residual analysis and mechanistic reasons were taken into consideration for regression selection (*Rockwell, Findlay & Cooke, 1987*; *Godfray, Partridge & Harvey, 1991*; *Davies, Krebs & West, 2012*).

Hatchling survival (*i.e.,* the probability of an individual hatchling to survive to fledging) was analysed in binomial generalized linear mixed-effects models (GLMM). The binary variable indicating fully-fledged goslings ($n = 619$) with a "1" and hatchlings that died ($n = 319$) with a "0" was used as binomial (Bernoulli) response variable. The ID of each pair and the year of observation (2004–2020) were again added as random effects. Six explanatory variables were added as fixed effects and were scaled before use in the GLMMs: (1) the local population size of Greylag Geese at the time of moult (MaxGrey); (2) the yearly number of breeding pairs of Greylag Geese (PairsGrey); (3) the highest weekly count per year of Egyptian geese, since they breed all year round (MaxEgypt); (4) the yearly number of breeding pairs of Egyptian Geese (PairsEgypt); (5) the initial number of hatchlings in each brood of Greylag Geese (Hatchlings); and (6) whether a pair dispersed with their young or not (Dispersal). Collinearity between variables in the models was taken into account, following the proposed threshold of <0.7 for the absolute value of correlation between variables proposed by *Dormann et al. (2013)*. All models which had strongly collinear variables as fixed effects were not considered for model selection.

Following the procedure described by *Symonds & Moussalli (2011)*, we used an IT-AIC all-subset approach for our model selection. We calculated the second order small sample corrected Akaike's information criterion (AICc) values and Evidence Ratio (ER) for every combination of our six explanatory variables and their interactions using the qpcR package (*Spiess, 2018*). We additionally calculated parameter estimates for the six models where the difference in the AICc value when compared to the best model ($\Delta$AICi) was lower than two to describe the effect of each variable on hatchling survival. A multimodel inference was performed by model averaging of these six models using the MuMIn package (*Bartón, 2020*). The conditional average was calculated for the models with a $\Delta$AICi lower than two (*Richards, 2005*).

## RESULTS

Since the first two Greylag Geese were seen in the Stuttgart area in 1981, their numbers reached a maximum of 342 individuals in 2015 but stabilized at 250 –350 geese since 2010 (Fig. 1A). A similar pattern of increase followed by stabilization was observed in the number of breeding pairs. Regular breeding started in 1995 and varied between one and three pairs until 2003. In 2004, numbers suddenly increased to eight and thereafter ranged between five and seven pairs until increasing again in 2009 to 14 breeding pairs. The number of breeding pairs ranged between 13 and 26 breeding pairs from 2009 to 2020 (Fig. 1B). Overall, the percentage of non-breeders in the population remained similar, ranging between 72% and 87% (Table S1).

In 2010, the first successful brood of Egyptian Geese occurred in Stuttgart. In just three years from 2016 to 2019 their numbers doubled from 86 to 174 individuals and in 2020, 230 Egyptian Geese were counted (Fig. 1A). Breeding pairs of Egyptian geese also increased to 14 pairs in 2017, briefly dropped to eight pairs in 2019 and rose again to 13 pairs in 2020 (Fig. 1B).

Over the years, the mean number of hatchlings per Greylag Goose pair ranged between 3.0 (2015) and 5.2 (2008), while mean fledgling numbers were between 1.2 (2015) and 4.7 (2008) (Fig. 1C). The annual fledging success (mean of all pairs) decreased initially from 100% in 2004 to a low of 38% in 2015, before increasing again to 62% in 2019 (Fig. 1D).

Between 2004 and 2020, a total of 104 different pairs of Greylag Geese bred in the study area, but only 61 pairs bred a second time, with subsequent broods per pair decreasing in number. Only ten pairs bred more than four times and of these, only two pairs bred a total of eight times each. Fledging success initially increased and was highest in pairs that bred for the 4th to 6th time. Later broods showed a decreasing fledging success per brood (Fig. 2A). However, when looking at the overall fledging success of pairs, classifying them by the total number of times a pair bred, the fledging success increased with the number of broods per pair (Fig. 2B).

Between 2004 and 2020 ($n = 237$ broods), Greylag Goose pairs had on average 3.96 hatchlings. Most pairs had between two and four hatchlings, but the optimal brood size regarding the proportion of surviving fledglings was skewed to larger brood sizes, which were rarer. Fledging success (in %) was highest in pairs with 11 hatchlings, followed equally by pairs with six and 15 hatchlings (Fig. 3).

In the GLMMs analysing the probability of a hatchling to fledge, some of the variables showed strong collinearity (Fig. S1). Using the IT-AIC all-subset approach for model selection, we selected a subset of all models (n = six out of 2099) in which the differences between the AICc value of the best model and the one of interest ($\Delta$AIC$i$) are less than two (Table 1). The best-ranked model, named M1, only includes three of the six variables as well as an interaction between two variables, indicating that these factors influence hatchling survival: local population size, number of hatchlings and dispersal. An increasing local population size had a strong negative effect on hatchling survival (Fig. 4A), while a higher number of hatchlings in a brood had a positive effect on their survival (Fig. 4B). Additionally, Greylag Geese which dispersed to the different brood rearing area had a
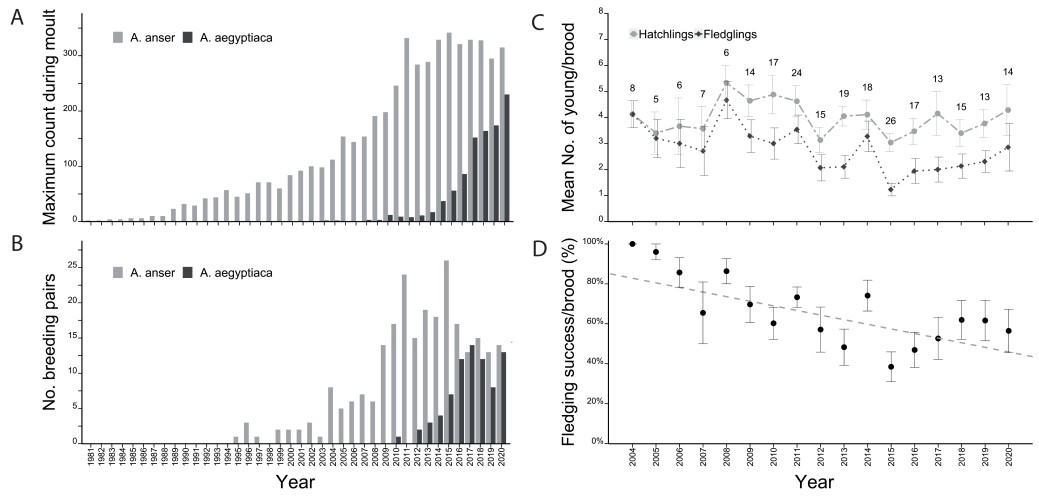

**Figure 1 Population development of Greylag and Egyptian Geese and fledging success of Greylag Geese.** (A) Maximum annual count and (B) number of breeding pairs of Greylag (light grey) and Egyptian Geese (dark grey) in the study area (1981–2020). Graphs adapted from *Hohmann & Woog (2021a)*. (C) The mean number of hatched (light grey dots) and fledged (black diamonds) goslings of Greylag Geese. (D) The fledging success per Greylag Goose brood (% of fledged/hatched). The annual mean fledging success per pair decreased over the years (lm: $y = -0.02x + 47.11$, $t_{0.01} = -4.18$, $p < 0.0001$).

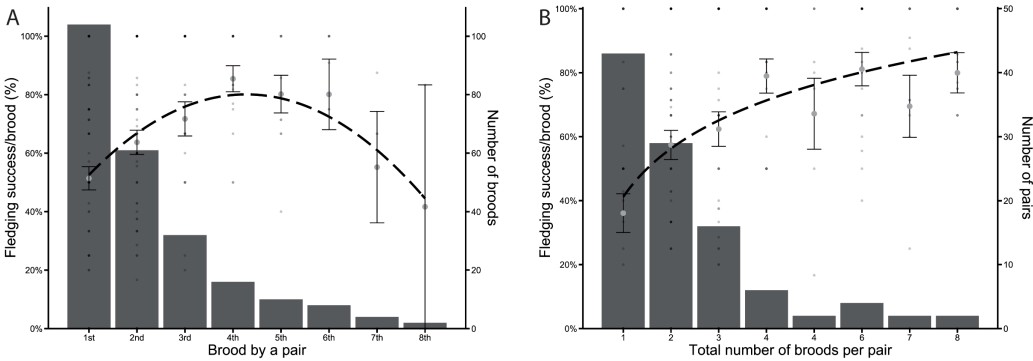

**Figure 2 Fledging success of individual pairs.** (A) Fledging success per brood (%) of the nth time a pair had a brood. The dashed line indicates a quadratic regression ($y = -2.57x2 + 21.95x + 33.18$). (B) Fledging success per brood (%) plotted against the total number of times each pair has bred. The dashed line indicates a logarithmic regression ($y = 21.75*\log(x) + 41.19$). Large grey dots in a, b indicate the mean of all fledging successes per brood $\pm$ SE, bars indicate number of broods (A) and number of pairs (B), respectively. The small dots represent each brood. Shading intensity increases with overlap with other broods.

higher hatchling survival (Fig. 4C), although the dispersal with large broods decreased hatchling survival. In 2009, we observed the dispersal of a pair with freshly hatched young for the first time. After hatching at the Max-Eyth Lake, they were resighted a day later in the Inner City Parks, at a distance of about 4 km along the river Neckar. Thereafter, up to five breeding pairs dispersed each year to the same location (Fig. S2). Across all years, a

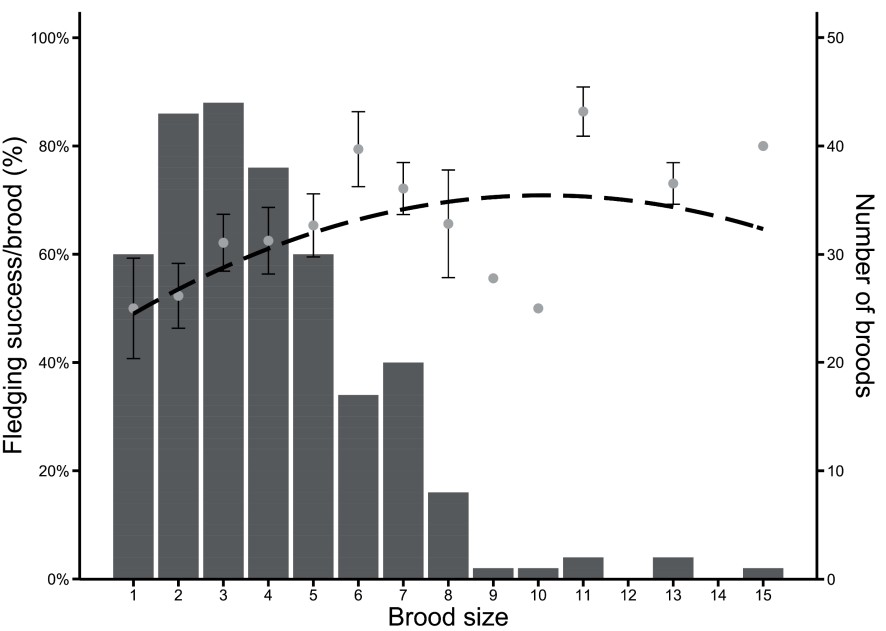

**Figure 3** **Frequency of brood sizes in Greylag Geese (2004–2020) in relation to the fledging success per brood (%).** The dashed line indicates a quadratic regression ($y = -0.26x^2 + 5.32x + 43.96$). Grey dots indicate mean fledging success $\pm$ SE and bars indicate number of broods.

**Table 1** **The six best-ranked models, for which ΔAICi ≤ 2.** Models 1-4 (bold) were used for conditional average in Table 3, grey shaded models contain variables with strong collinearity and were disregarded for model selection.

| M | Variables | k | AICc | ΔAICi | ER |
|---|---|---|---|---|---|
| **M1** | MaxGrey + Hatchlings + Dispersal + Dispersal*Hatchlings | 7 | 630.3565 | 0 | 1 |
| **M2** | PairsGrey + PairsEgypt + Hatchlings + Dispersal + Dispersal*Hatchlings | 8 | 630.8385 | 0.4820 | 1.2725 |
| **M3** | MaxGrey + Hatchlings + Dispersal + Dispersal*Hatchlings + Dispersal*MaxGrey | 8 | 632.1746 | 1.8181 | 2.4820 |
| **M4** | MaxGrey + Hatchlings + Dispersal | 6 | 632.2186 | 1.8622 | 2.5372 |
| M5 | MaxGrey + PairsGrey + Hatchlings + Dispersal + Dispersal*Hatchlings | 8 | 632.2309 | 1.8744 | 2.5528 |
| M6 | MaxGrey + PairsEgypt + Hatchlings + Dispersal + Dispersal*Hatchlings | 8 | 632.2731 | 1.9166 | 2.6073 |

**Notes.**

M, model name; k, number of fitted parameters; AICc, small-sample corrected Akaike's information criterion; ΔAICi, difference in AICc value with the best model; ER, Evidence Ratio.

Greylag Geese: MaxGrey, maximum annual count during moult; PairsGrey, number of breeding pairs; Hatchlings, number of hatchlings per brood; Dispersal, whether a pair dispersed with their young or not (yes/no).

Egyptian Geese: MaxEgypt, maximum annual count; PairsEgypt, number of breeding pairs.

Interactions between variables indicated by * (Dispersal*Hatchlings and Dispersal*MaxGrey).

total of 13 different breeding pairs dispersed, some of which did so repeatedly in several years. From 2016 onwards, fewer geese dispersed, ranging between one and three pairs each year.

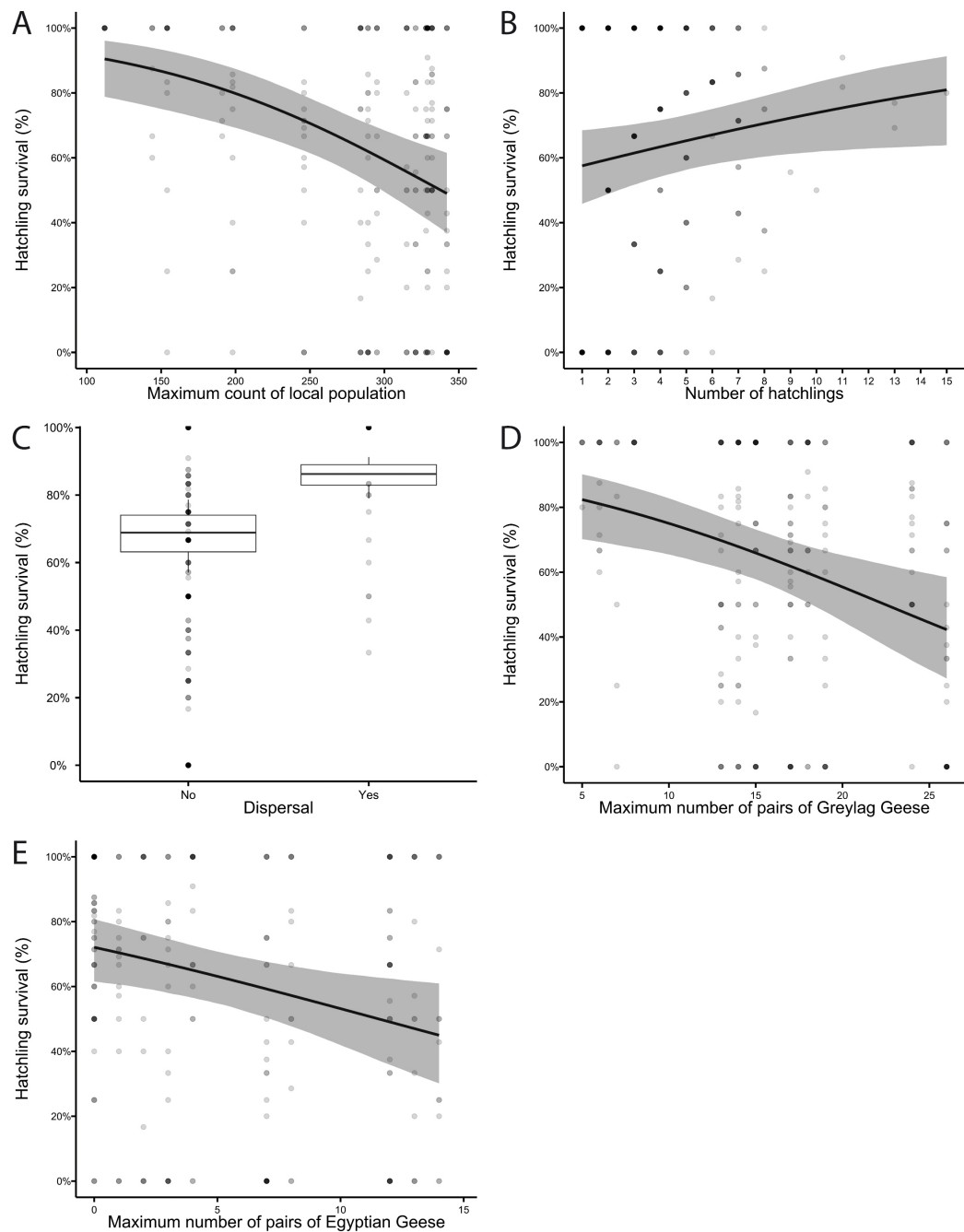

**Figure 4** **The probability of a hatchling to fledge (hatchling survival (%)) (A) decreased with the maximum annual count of the local population and (B) increased with the number of hatchlings per pair. (C) Dispersed pairs ($n = 30$) had a higher hatchling survival than pairs that did not ($n = 207$). Hatchling survival (%) was negatively affected by (D) the number of Greylag Geese breeding pairs and (E) the number of Egyptian Geese breeding pairs.** Graphs from model M1 (A–C) and M2 (D, E). Each hatchling is represented by a small dot; shading intensity of small dots increases with the overlap with other hatchlings thus darker dots indicate a higher number of hatchlings. Shaded area in A, B, D, E represents the 95% confidence interval (CI).

**Table 2 Binomial generalized linear mixed-effects models of the two best-ranked models M1 and M2.** Models show the effects of the local population size of Greylag Geese at the time of moult (MaxGrey), the yearly number of breeding pairs of Greylag Geese (PairsGrey), the yearly number of breeding pairs of Egyptian Geese (PairsEgypt), the initial number of hatchlings in each brood of Greylag Geese (Hatchlings) and whether a pair dispersed with their young or not (Dispersal). CI, 95% confidence interval.

| | | | | Hatchling surival (%) | | |
|---|---|---|---|---|---|---|
| *Model* | **M1** | | | **M2** | | |
| **Fixed Effects** | | | | | | |
| *Variables* | *Estimate* | *CI (95%)* | *p* | *Estimate* | *CI (95%)* | *p* |
| (Intercept) | 0.55 | 0.16–0.93 | **0.006** | 0.53 | 0.18–0.88 | **0.003** |
| MaxGrey | −0.65 | −1.01 to −0.30 | **<0.001** | | | |
| Hatchlings | 0.19 | 0.01–0.38 | **0.043** | 0.19 | 0.01–0.38 | **0.044** |
| Dispersal | 0.69 | 0.39–0.99 | **<0.001** | 0.68 | 0.38–0.98 | **<0.001** |
| Dispersal * Hatchlings | −0.30 | −0.61–0 | **0.048** | −0.31 | −0.61 to −0.01 | **0.044** |
| PairsGrey | | | | −0.50 | −0.83 to −0.18 | **0.002** |
| PairsEgypt | | | | −0.41 | −0.73 to −0.10 | **0.010** |
| **Random Effects** | **M1** | | | **M2** | | |
| $N_{ID}$ | 104 | | | 104 | | |
| $\tau_{00\,ID}$ | 0.76 | | | 0.73 | | |
| $N_{YR}$ | 17 | | | 17 | | |
| $\tau_{00\,YR}$ | 0.30 | | | 0.21 | | |
| Observations | 938 | | | 938 | | |
| Marginal $R^2$/ Conditional $R^2$ | 0.178/ 0.379 | | | 0.186/ 0.367 | | |

As the number of breeding pairs of both species was not included in M1, we used model M2, the second-best model, to analyse their effect. Regardless of species, the increasing numbers of breeding pairs had a negative effect on Greylag Geese hatchling survival (Figs. 4D, 4E).

Dispersal and the size of the local Greylag Goose population had the strongest effect on hatchling survival, followed by the number of hatchlings per pair. The interaction between dispersal and the number of hatchlings was also represented in the best-ranked model, with a negative interaction (Tables 2, 3, Table S2). Of the six variables, the maximum count of Egyptian Geese was not included in the conditional average as it does not appear in the best-ranked models.

## DISCUSSION

In the newly colonised area in Stuttgart, the number of Greylag Goose breeding pairs increased with the total population size. Subsequently, the fledging success of individual goose pairs decreased, indicating density dependence, possibly through competition for resources or nesting space. A similar observation was made in the Svalbard Barnacle Goose population, though no significant density dependence effects on reproductive success (fledgling survival) could be shown there. Instead, the total population size negatively influenced population growth (*Loonen, Tombre & Mehlum, 1998*; *Layton-Matthews et al.,*

**Table 3  Conditional averages of hatchling survival of the four non-collinear models with ΔAIC*i* < 2.**

| Variable | Estimate | Std. Error | z-value | p-value |
|---|---|---|---|---|
| (Intercept) | 0.54 | 0.19 | 2.78 | 0.005 |
| MaxGrey | −0.64 | 0.18 | 3.45 | <0.001 |
| PairsGrey | −0.50 | 0.17 | 3.01 | 0.003 |
| Dispersal | 0.67 | 0.16 | 4.31 | <0.001 |
| Hatchlings | 0.21 | 0.10 | 2.09 | 0.037 |
| Dispersal * Hatchlings | −0.30 | 0.15 | 1.94 | 0.053 |
| Dispersal * MaxGrey | 0.11 | 0.20 | 0.58 | 0.565 |
| PairsEgypt | −0.41 | 0.16 | 2.55 | 0.011 |

**Notes.**
Greylag Geese: MaxGrey, maximum annual count during moult; PairsGrey, number of breeding pairs; Dispersal, whether a pair dispersed with their young or not (yes/no); Hatchlings, number of hatchlings per brood. Egyptian Geese: PairsEgypt, number of breeding pairs.

*2019*; *Layton-Matthews et al., 2020*). It may be possible that such a situation occurred in Stuttgart, with the local population size as a possible driving factor behind the stagnation of population growth. However, as we measured no ecological parameters, we cannot disregard their effect on fledging success. Direct density-dependent effects on fledging survival are also possible, as has been shown in Lesser Snow Geese (*Cooch et al., 1989*) and Black Brant Geese *Branta bernicla nigricans* (*Sedinger et al., 1998*). Studies in other bird species have also shown that fledgling mortality increases with increased population density (*e.g.*, Grey Partridge *Perdix perdix* in *Bro et al., 2003* or Savannah Sparrows *Passerculus sandwichensis* in *Woodworth et al., 2017*).

When a local population reaches a high density in a defined space, there are more individuals that have to share the same quantity of resources such as food or nesting places. Crowding of animals may lead to social stress, as, given more room, individuals would space themselves more generously (*Metcalf, Hampson & Koons, 2007*). Since nesting sites at the Max-Eyth Lake are limited, breeding pair numbers stagnated since 2009. This aligns with known data from other goose species, such as Barnacle Geese in Svalbard (*Owen, 1984*; *Prop, van Eerden & Drent, 1984*; *Black, 1998*). Likewise, in migratory Dark-bellied Brant Geese *Branta bernicla bernicla*, the limiting factor regarding breeding was the availability of safe nesting sites (*Ebbinge et al., 2002*). In the Svalbard population of Barnacle Geese, the recruitment rate declined with increasing population size (*Owen & Norderhaug, 1977*), which increased the proportion of non-breeders in the population (*Owen, 1984*; *Black, Prop & Larsson, 2007*). In contrast, in our local study population, the proportion of breeders to non-breeders remained similar. This is reflected in the stabilization of both the local population size (as seen from the maximum count) and the number of breeding pairs, which occurred around the same time (2010). It appears likely that some juvenile Greylag Geese will leave the study area, which is supported by resightings of Stuttgart Greylag Geese in the surrounding areas (pers. obs. H. Haag, 2018). Further studies are needed to explore the dispersal of the Stuttgart Greylag Geese to the surrounding areas and bring them into context with our results.

Fledging success was highest between the 4th and 6th brood of a pair and decreased again when a pair bred together more than seven times, indicating a trade-off between experience and age. Experienced pairs will have an advantage over first-time breeders, but reproductive success will decrease with age (*Fowler, 1995*). While we do not know the ages of many of our breeding pairs, reproductive success in Bar-headed Geese *Anser indicus* has been shown to be influenced by the ages and the experience of the partners (*Lamprecht, 1990*). The initial increase in fledging success in our study indicates that pairs which had already bred together were more successful. In Lesser Snow Geese, more successful pairs were more attentive to their young (*Cooke, Bousfield & Sadura, 1981*), while in Barnacle Geese, pairs which were successful in one year often continued to have higher reproductive success in the following year (*Raveling, 1981*). As only 61 of our Greylag Goose pairs bred more than once and only 10 pairs more than five times, competition regarding nest sites on the breeding islands appeared to be high. This is also reflected in the stagnating number of breeding pairs.

The overall fledging success of the individual pairs was higher for those pairs which bred more often in total. Even though fledging success decreased in later broods, breeding more often was still advantageous for successful pairs overall. It is unknown why certain pairs bred more often than others, though there may be inherent qualities leading to a higher number of broods and fledging success. A higher dominance rank may be useful in retaining a nest site throughout the years. While we have not analysed social structure in regard to dominance, in Svalbard Barnacle Geese, families with young win more encounters than pairs or singles, indicating a higher dominance rank (*Black & Owen, 1989*). Recent studies have also shown that there may be an association between reproductive success and heterozygosity (*Seddon et al., 2004*), which may be an interesting future research question to address in our study population.

As the breeding islands are inaccessible, we were unable to determine the clutch sizes. Our annual mean number of hatchlings per pair varied between three and five young, which is similar to the mean clutch sizes of other populations of Greylag Geese (*Nilsson & Persson, 1994*). Three hatchlings per pair were most common, while fledging success was highest in pairs with six hatchlings (except for some exceptionally large broods of 11 or 15 hatchlings). While large brood sizes are uncommon (*Glutz von Blotzheim, 1990*), there may be an advantage in raising large broods. In Barnacle Geese, families with large broods have a higher social rank than families with smaller broods and profit from this by better grazing opportunities (*Black & Owen, 1989*). In wild Greylag Geese from Sweden, families with more than three goslings also had a higher reproductive success (survival from hatchling to fledgling) than families with less than three (*Nilsson & Persson, 1994*). This mismatch between most frequent and most successful brood size may indicate resource competition prior to or during egg laying. Contrary to migratory geese, resident geese acquire fat in close vicinity to the nesting site. Alternatively, as a long-lived species, Greylag Geese may possibly reduce their brood size to maximize their own survival into the next year. In Common Eiders *Somateria mollissima*, females adjust laying date and clutch size based on both external (*i.e.,* environmental) and internal (*i.e.,* body condition) influences (*Descamps et al., 2011*). Most parents would produce clutch sizes which are smaller than physically

possible, and as parents have different abilities and experiences, the optimal brood size will vary strongly among pairs (*Drent & Daan, 1980*).

Using GLMMs, we measured the influence of a total of six variables on the likelihood of a hatchling to fledge, *i.e.,* hatchling survival. Some of our variables showed strong collinearity: the maximum count of Greylag Geese was collinear with the number of Greylag Goose breeding pairs. Likewise, the maximum count of Egyptian Geese was collinear with the number of Egyptian Goose breeding pairs. While we included them as variables in our analysis, models containing these collinear variables should be interpreted with caution. We focus on the two best-ranked regression models, which do not contain collinear variables. However, the remaining models among the six best-ranked regression models cannot be disregarded, since ΔAIC*i* is less than two (*Richards, 2005*). Overall, we find that dispersal had the highest positive effect on hatchling survival, while the maximum count of Greylag Geese had the highest negative effect. The number of hatchlings influenced hatchling survival both in combination with dispersal and on its own. Higher numbers of hatchlings per brood generally positively influenced hatchling survival, except when the brood dispersed. Then, higher numbers were a detriment to hatchling survival. The three variables and the interaction were also the variables remaining in the best-ranked model M1. As the maximum count of geese led to a decrease in the probability of hatchling survival, density dependent effects are likely. The results of the GLMM support our individual analyses on these variables.

The higher hatchling survival of dispersed pairs is surprising, as the journey between the nesting and the brood rearing area is hazardous. The families have to cross the main four-lane traffic route as well as tram tracks on the last stretch. In an evolutionary sense, dispersing birds were more successful and this could explain the observed behaviour. As we did not measure food quality, predation pressure, or social stress we can only speculate why dispersal was a success. Initially, with increasing population size, feeding competition between families may have increased near the original nesting sites, as has been shown for example in Barnacle Geese (*Owen, 1984*). Many goose species react to poor feeding conditions by moving away from the breeding areas (*Cooch et al., 1993*) and indeed, geese are known to select habitats which will optimize their reproductive success (*Cody, 1985*). Since goose families within an area usually form crèches, local dominance of some breeding pairs seems an unlikely explanation for the observed dispersal. As dispersal leads to a higher fledging success, it is perhaps surprising that it remains a rare event. The number of dispersing pairs per year decreased after 2016, which coincided with heavy construction work at the described passage (*Landeshauptstadt Stuttgart, 2021*). In the future, in-depth field observations of the breeding area during hatching time are necessary to explore why some pairs choose to disperse yet others do not.

The increasing local population size of Egyptian Geese had no apparent influence on Greylag Goose fledging success, but this may change in coming years. The number of Egyptian Geese breeding pairs already had a negative effect. Introduced bird species may have a negative impact on the existing avifauna, although there are few studied examples. Among the most common interactions are competition for nesting sites and food (*Baker, Harvey & French, 2014*). While Egyptian Geese are known to displace other birds from

nesting sites, this has not occurred at the Max-Eyth Lake (*Hohmann & Woog, 2021b*). Competition for food in our study area seems unlikely, as grazing is available throughout the year, albeit in varying quality (*Käßmann & Woog, 2007*). Introduced bird species may also have a positive impact on native species, for example through facilitation by preferential predation (*Roemer, Donlan & Courchamp, 2002*). As we have no data on predation, we can only speculate if Egyptian Geese serve such a role for Greylag Geese. While Egyptian Geese are more vigilant than Greylag Geese (*Hohmann & Woog, 2021a*), they may still have a dilution effect by increasing the number of available prey (*Duca, Brunelli & Doherty Jr, 2019*).

As we study an urban population of Greylag Geese, we cannot disregard the possible influence of urbanisation on reproductive success. While several studies have looked at urbanisation as a potential factor affecting reproductive success, results are inconclusive (*Chamberlain et al., 2009*). In some studies, urban birds have been shown to have poor reproductive success, particularly when compared to rural populations of the same species (*e.g.*, Great Tits *Parus major* in *De Satgé et al., 2019*), while in others, more fledglings were produced in suburban or urban landscapes (*e.g.*, House Wren *Troglodytes aedon* in *Newhouse, Marra & Johnson, 2008* and Northern Raven *Corvus corax* in *Kristan & Boarman, 2007*). Greylag Geese have recently started to establish local breeding populations in the rural areas beyond Stuttgart, which will allow a comparison of reproductive success along an urban-rural gradient in the future.

## SUMMARY AND CONCLUSION

In an urban local population of Greylag Geese, we were able to show that the dispersal of pairs from the nesting site to a different brood rearing area appeared advantageous as it resulted in a higher fledging success. Additionally, a higher number of hatchlings generally resulted in higher fledging success. However, the increasing population size had a negative impact on fledging success, suggesting density dependence. Pairs which were able to breed together more often showed a higher fledging success. Our results show that some newly established populations in urban settings do not grow indefinitely, which is an important fact that should be taken into account by wildlife managers.

## ACKNOWLEDGEMENTS

We thank the many volunteers who assisted with the yearly ringing and the birders of the area who provided us with opportunistic sightings of ringed geese. We are grateful to Dr. Fränzi Korner-Nievergelt for statistical advice and Jonah Ulmer for commenting on an earlier draft of this manuscript. We thank the editor and two anonymous reviewers for their valuable comments on the manuscript.

### Funding
Goose monitoring was funded by the Ministerium für Ernährung, Ländlichen Raum und Verbraucherschutz Baden-Württemberg (AZ 31-0826.54774 (2007-2021)). The funders had no role in study design, data collection and analysis, decision to publish, or preparation of the manuscript.

### Grant Disclosures
The following grant information was disclosed by the authors:
The Ministerium für Ernährung, Ländlichen Raum und Verbraucherschutz Baden-Württemberg (AZ 31-0826.54774 (2007-2021)).

### Competing Interests
The authors declare there are no competing interests.

### Author Contributions
- Sabrina Mai conceived and designed the experiments, analyzed the data, prepared figures and/or tables, authored or reviewed drafts of the article, and approved the final draft.
- Jean-Louis Berthoud analyzed the data, prepared figures and/or tables, authored or reviewed drafts of the article, and approved the final draft.
- Holger Haag performed the experiments, authored or reviewed drafts of the article, and approved the final draft.
- Friederike Woog conceived and designed the experiments, analyzed the data, authored or reviewed drafts of the article, and approved the final draft.

### Animal Ethics
The following information was supplied relating to ethical approvals (i.e., approving body and any reference numbers):
Permissions to catch and ring geese were obtained from the Regierungspräsidien Stuttgart and Karlsruhe (AZ 55-8853.17/S; 55-9213.47 and 55-8841.03).

### Data Availability
The raw data and the corresponding R script are available in the Supplemental Files.

### Supplemental Information
Supplemental information for this article can be found online at http://dx.doi.org/10.7717/peerj.13685#supplemental-information.

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
