# Peer review of "Factors limiting reproductive success in urban Greylag Geese (Anser anser)"

_PeerJ, doi:10.7717/peerj.13685_

## Round 0.1 · original submission · Major Revisions

Dear authors, liebe Frau Woog

As you can see, our reviewers raised a number of issues that you should address in your revision. As the reviewers will see your revision, make sure that you follow their recommendations.

Kind regards
Michael Wink

Reviewer 1 ·

Basic reporting

no comment

Experimental design

Line 104: about the hypotheses, this section of the manuscript should be deeply rethink/reshaped. The referred hypotheses should be better stated as questions/tasks or points guiding the manuscript methodological design.

(1) In the newly colonized area, an increasing population size of Greylag Geese leads to an increase in the number of breeding pairs as more offspring can be recruited into the breeding population.

There are two main factor influencing population growth (total number, be aware of the effective population size that is what actually represents the reproductive potential of the demo):
i) Natural increment of the population by the increment of the number pairs
ii) inmigration
If the population is growing but not the number of breeding pairs then the increase in the population is due to inmigration, birds than for some reasons do not breed, is it not happening? How an increment in the population size cannot lead to the increment of the breeding pairs and the total population size? The relationship cause-consequence here is not clear and it should be clarified.

(2) Fledging success of individual goose pairs decreases due to density dependence.
Since you are not measuring any other ecological parameter than number of birds you can find a correlation between fledging success and density but you cannot discard other factors acting against individual goose pairs fledging success.
How does this hypothesis relate/contradict with point (1)?

(3) Pairs that emigrate with young goslings to a new brood rearing areas will have a lower fledging success because they are more likely to lose young during the journey.

This can be tested as a hypothesis. Nevertheless, these movements are happening in about 4 Km from the breeding site an in the text it is referred as an emigration. I think the concepts of local displacements/emigration etc should be reviewed carefully and use the appropriate one. Since the movements of some pairs to other near areas seems not to be dangerous, then is there an alternative route to get to the new location?

(4) There is an optimal brood size which yields the highest fledging success.
This statement is tautologic, since there is always an "optimal" brood size (or range) which lead to the highest fledging success.

(5) On an individual level, the probability of a hatchling to fledge decreases with the following variables or a combination thereof:
a) an increasing local population size;
b) the number of breeding pairs in a population

(is it independent from a?)

c) emigration (see hypothesis 3)

Is not redundant to question 3?

d) a higher number of hatchlings
e) the increasing number of Egyptian Geese due to competition or agonistic behaviour. Regarding individual pairs, fledging success

These points are an extension of the question No 2.

(6) will increase with breeding experience and

This statement is tautologic, do not reproductive success always correlate with parents experience?

(7) will be overall higher for pairs which bred more often
This is related to point 6.

Validity of the findings

Figure 3 b,d and e are not convincing, even though the adjusted line shows a tendency the conclusions should be taken carefully since the visual inspection of the graph show a cloud-like dispersion of the observations. It would benefit if the dots are presented in a darker colour along with the corresponding regression coefficients.

Reviewer 2 ·

Basic reporting

no comment

Experimental design

no comment

Validity of the findings

no comment

Additional comments

This is a paper describing reproductive success in a population of urban greylag geese in the city of Stuttgart, where individuals have been monitored since the establishment. This long-term dataset (17 years) provides a good opportunity to understand processes in an increasing goose population and the related reproductive success of individuals and possible impacting factors.

Some general comments:

o I’m not familiar with the format of this journal, but it looks a bit strange with figure legend in-between the written text. Fine if this is the standard.

o The Introduction will be improved from some re-structuring. E.g., a detailed description of barnacle geese is given, but the full story is not completed but continues with information from the snow goose. I suggest that the authors more specifically list the various possible factors affecting reproductive success (RS) in more general terms regarding birds, and then provides some examples where geese are the main examples.

o I think the Discussion is too long, with too many references to other bird species than geese. I acknowledge the authors overview in the field, but some of it appears not directly relevant. However, I leave for the Editor to judge the length of the Discussion in light of the journal style. If it is OK, then fine with me!


Specific comment referring to lines in the document:

Lines 28-29: Add that also the adult survival is high?

Line 35: “regulation” instead of “management” ?

Lines 36-38: write more precisely that the shift to agriculture is a reason for population increase (shift the order of the statements)

Lines 46-47: Describe briefly the mechanisms for density dependence in this case

Lines 47-49: How? (effect on RS)

Line 54: add “..of the annual cycle and locations as well”.

Lines 58-60: There is nothing about hunting in the Owen & Black paper on barnacle geese? I think the Danish group lead by Jesper Madsen may have some better and alternative references on pink-footed geese.

Line 63: use “estimated” or “counted” for the 300 individuals in Caerlaverock, as the 2nd WW apparently also had an effect on where the geese stayed during these years.

Line 64: slower rate; what are the numbers?

Lines 84-85: present what the arguments for the optimal clutch size are

Line 101: …fledged in the same area, adding…

Lines 105-107: other way around? More recruited leads to population increase.

Line 107: …individual goose pairs decrease over time due to…

Line 110: (4) explain what the logic is behind this

Line 112: …thereof due to density dependence mechanisms…

Lines 114-116: use the same format at in (1)-(5)

Line 121: do the urban parks also include ponds and open water?

Line 124: last sentence, it is obvious, delete?

Line 130: 300 pairs, from how many individual originally? What is known?

Line 131: how large proportion?

Line 137: the predators – both raptors and terrestrial?

Line 139: ….leading to higher nest densities.

Line 140: when disperse – winter/summer?

Line 144: use “introduced” instead of “occurred”?

Line 147: add average clutch sizes

Line 149:…breeding advantage due to diverse nesting site preferences.

Line 152: use “as a consequence of” instead of “during”

Line 162: …highest estimate based on goose counts during a given year….

Line 164: …Directive and also used by…

Line 168: How long is the narrow time period?

Line 177: Somewhat unclear what the local population is – define…. What if some geese were breeding elsewhere not detected in the study? Maybe add some distance information to clarify and define what the local population size is?

Line 183: adjust the text as the real number of eggs hatched was not possible to count as there were no entrance on the breeding island.

Line 185: define period for survival to fledging

Lines 201-207: the abbreviation of concepts, understand the need to shorten for the tables etc. (model presentations), but one word would be much better – e.g. HA= hatchlings, etc. They can still be defined in the text, but this will be easier for the general reader to follow and understand which variables are included in the different models.

Line 222: How many geese were observed in 1988?

Lines 230-231: …three years from 2016 to 2019 their numbers doubled….

Line 232: delete “a high of”…

Lines 269-273: say something about the large brood sizes with hatchlings that are well above the normal clutch size.

Lines 277-283: some of this is already said above, delete and add “strong” etc. in the results described above instead, for a more streamlined text.

Lines 313-314: describe the mechanisms behind this statement

Line 317: see also some of the work from the Dutch team on Svalbard (1998, Nor. Polar Skrifter), may be information also referred to at this point.

Lines 317-319: add a question mark or re-write

Lines 331-333: see also the Black paper from 1998 about the colonization process on Svalbard for barnacle geese.

Line 359: but what is the detectability for families dispersing?

Line 367: add why so large broods

Lines 371-372: but geese also use body reserves for egg-development, reserves that may be gained elsewhere. Explain.

Lines 392-393: delete the parenthesis defining the M1 model, or say with words

Lines 405-410: but what about the dilution effect for the broods when more goose families are present, could that have an effect?

Line 412: use “larger” instead of “more”

Line 415: …indicus have been shown to be influenced by the ages…

Conclusion: most of this is more like a summary rather than a conclusion. The authors should have another look at this section.

Line 450. Use “suggesting” instead of “hinting towards”?

Line 452: Difficult to predict generally for all colonies – this is one case only. What about the exponential increase of barnacle geese in e.g. Malmö? No need to describe that example, but be more careful to suggest generality based on one case only.

---

## Round 0.2 · Minor Revisions

Dear authors

Still a minor revision is required.

Greetings
M. Wink
Academic Editor

Reviewer 1 ·

Basic reporting

The authors have considered the main points raised in the first round of review. Nevertheless, in my opinion, some important aspects still need consideration before recommending the manuscript for publication.
The manuscript would benefit of a couple of sentences clearly explaining the benefit of using fledging success and hatchling survival, the different information they provide separately and the things the readers need to be aware at the time to interpret the results. As I understand it, the complexity of the models (number of variables) behind the estimation of these indexes make the difference, right?
I have to stress the point of referring to the movements 4 km away from the Max-Eyth Lake emigration and not dispersion. Since birds return to their original breeding place the movement is not an emigration, "Across all years, a total of 13 different breeding pairs emigrated, some of which emigrated repeatedly in several years. From 2016 onwards, fewer geese emigrated, ranging between one and three pairs each year." I think the authors need to go over the concepts and refer to what they are describing in the manuscript correctly.
Regarding the Figure 3 (panels b and d) I would still suggest to use different colours for a range of number of hatchlings, using shading intensity is misleading and create a bias in the data interpretation; this information should be clearly stated in the figure caption.

Experimental design

'no comment'

Validity of the findings

'no comment'

Additional comments

'no comment'

---

## Round 0.3 · accepted · Accept

Dear authors

Thank you for adequately responding to the recommendations of the reviewers. Your ms is now ready and I can accept it for publication.

Congratulations
Kind regards
Michael Wink
Academic Editor